# Risk Analysis for Short-Term Operation of the Power Generation in Cascade Reservoirs Considering Multivariate Reservoir Inflow Forecast Errors

**Yueqiu Wu [1], Liping Wang [1], Yi Wang [1],\*, Yanke Zhang [1], Jiajie Wu [1], Qiumei Ma [1], Xiaoqing Liang [2] and Bin He [3]**

[1] School of Water Resources and Hydropower Engineering, North China Electric Power University, Beijing 102206, China; wuyueqiu@ncepu.edu.cn (Y.W.); lpwang@ncepu.edu.cn (L.W.); ykzhang2008@163.com (Y.Z.); 1172111017@ncepu.edu.cn (J.W.); qiumeima@ncepu.edu.cn (Q.M.)

[2] Department of Physics and Hydropower Engineering, Gansu Normal College for Nationalities, Hezuo, Gansu 747000, China; l_x_q1688@163.com

[3] National-Regional Joint Engineering Research Center for Soil Pollution Control and Remediation in South China, Guangdong Key Laboratory of Integrated Agro-Environmental Pollution Control and Management, Institute of Eco-Environmental and Soil Sciences, Guangdong Academy of Sciences, Guangzhou 510650, China; bhe@soil.gd.cn

\* Correspondence: hywy02@foxmail.com; Tel.: +86-15201173031

**Abstract:** In the short-term operation of the power generation of cascade reservoirs, uncertainty factors such as inflow forecast errors could cause various types of risks. The inflow to a downstream reservoir is not only affected by inflow forecast errors from upstream reservoirs but also the forecast errors associated with inflow to the stream segment between the reservoirs, such as from a tributary. The inflow forecast errors of different forecast periods may also be correlated. To address this multivariate problem, the inflow forecast error variables were jointly fitted in this study using the Gaussian mixture model (GMM) and a *t*-Copula function based on the analysis of the error distribution characteristics in different forecast periods. Therefore, a stochastic model that coupled with the GMM and *t*-Copula to calculate inflow forecast errors in multiple forecast periods was established. Furthermore, according to the simulation results of the stochastic model and the predicted runoff series, a set of simulated runoff processes were obtained. Then they were combined with the existing power generation plan to carry out the risk analysis for short-term operation of the power generation in a cascade reservoir. The approach was evaluated using the Jinguan cascade hydropower system within the Yalong River basin as a case study. For this case study, the risk analysis for short-term operation of the power generation was analyzed based on stochastic simulation of the inflow forecast errors; the results show the feasibility and effectiveness of the proposed methods.

**Keywords:** cascade reservoirs; short-term operation of the power generation; risk analysis; multivariate analysis; inflow forecast errors

## 1. Introduction

The accuracy of inflow forecasts is crucial to the formulation of plans for short-term operation of the power generation from cascade reservoirs [1,2]. Although hydrologic forecast techniques have marked improvement in theory and practice in recent years [3–5], forecast errors remain inevitable. Forecast inflow is often directly used to formulate operation schemes for short-term operation of the power generation in cascade reservoirs. Forecasted inflow does not currently consider the influence of various uncertainty factors which may cause risks of insufficient output, wasted water and beyond-or-below-reservoir limit water levels, because of deviations between the actual and predicted inflow. As a result, the optimal operation schemes cannot be currently applied directly to actual power generation processes [6]. Thus, the risk analysis for short-term operation of the power generation in complex reservoirs is needed to reduce power and economic losses.

The risk factors affecting reservoir operation include hydrologic, hydraulic and engineering factors [7–10]. In recent years, scholars have conducted research on the reservoir operation risk of these factors. Hydrologic factors mainly concern the uncertainty of hydrologic phenomena and hydrologic models [11–13]. Historically, the consideration of inflow forecast errors mostly focused on stochastic simulations [14,15] and random quantification [16–22]; some scholars used fuzzy theory to deal with inflow forecast errors [23]. For example, to analyze the influence of forecast errors for different forecast periods on runoff process forecast errors, Ji et al. [24] established a stochastic simulation model of inflow forecast errors for multiple forecast periods for a single reservoir by using a meta-student *t*-Copula function. An improved Gaussian mixture distribution and Markov chain Monte Carlo algorithm were constructed to model the measured forecast errors and generate ensemble inflow forecasts [25]. Most of these studies focus on a single forecast period; they have not considered the correlation between different forecast periods, which reduces the simulation accuracy of inflow forecast errors by calculating multiple errors.

However, for cascade reservoirs, which are usually placed in basins with many tributaries, the factors that influence downstream reservoir inflow include not only the forecast errors of the main stream, but also the forecast errors of inflow from tributaries between the reservoirs. Thus, a comprehensive analysis of the correlation between each forecast period and runoff process is needed in order to effectively handle the risk of power generation operation and enhance the benefits of cascade reservoirs.

Previous studies mostly considered the inflow forecast errors of a single reservoir and rarely considered the correlation of forecast errors between different sources of runoff processes [23–25]. Few studies have examined the risk of power generation operation of cascade reservoirs caused by forecast errors of different runoff processes [23–26]. Therefore, this paper proposes a novel approach to consider the correlation between all forecast errors in the forecast period. First, the forecast error variable which was described by the Gaussian mixture model (GMM) function of each runoff process during each forecast period was obtained using a statistical method based on historical (observed) and forecast runoff data. Then, by analyzing the correlation among the forecast error variables, the joint function of multivariate inflow forecast errors was developed using the *t*-Copula method. Finally, runoff was simulated by the Monte Carlo method to analyze the risk for short-term operation of the power generation in cascade reservoirs.

## 2. Materials and Methods

In the operation of cascade reservoirs, various tributaries may flow into the basin, and forecast errors could occur and vary in the runoff forecasting of each tributary in different forecast periods. Some of these forecast errors may have correlations with each other while others may not have any correlations. The consideration of their correlation has a significant impact on the accuracy of inflow forecast errors modeling. The analysis of forecast errors proposed here is described below.

### 2.1. Multivariate Inflow Forecast Errors

As shown in Figure 1, for large cascade reservoirs, we assume that the main stream is $A$, with $n$ tributaries ($B_1$, $B_2$, ... , $B_n$) and $k$ reservoirs. There are $T$ forecast periods and $M$ sets of forecast inflow.

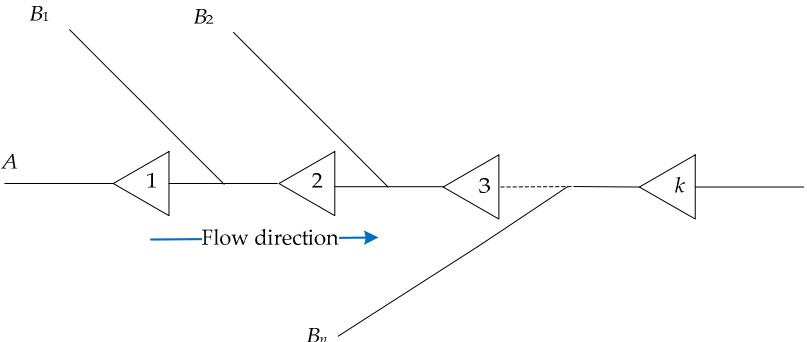

**Figure 1.** Cascade reservoirs in a basin. In order to make the calculation easy to be understood, the interval branch is simplified as one.

The inflow forecast errors are in the form of a relative value, as shown by:

$$X^j = \frac{Q^j_{forecast} - Q^j_{actual}}{Q^j_{actual}} \times 100\% \tag{1}$$

where $X^j$, $Q^j_{forecast}$ and $Q^j_{actual}$ represent the relative inflow forecast errors, the forecast inflow value and the actual inflow value during the forecast period $j$ ($j = 1, 2, \ldots,$ $J$), respectively.

The simulated inflow is shown as:

$$Q^j_{simulated} = \frac{Q^j_{forecast}}{1 + X^j_{simulated}} \tag{2}$$

where $Q^j_{simulated}$ is the simulated inflow, and $X^j_{simulated}$ is the simulated inflow forecast error.

Then based on simulated inflow forecast errors, simulated inflow can be obtained by Equation (2).

If the forecast error vector of the main stream $A$ under one set of runoff data is expressed as $(X^1, X^2, \ldots, X^J)$, then the following forecast error matrix can be established under $M$ sets of historical runoff data, such that:

$$X = \begin{bmatrix} x_1^1 & x_1^2 & \cdots & x_1^J \\ x_2^1 & x_2^2 & \cdots & x_2^J \\ \vdots & \vdots & \vdots & \vdots \\ x_M^1 & x_M^2 & \cdots & x_M^J \end{bmatrix} \tag{3}$$

The forecast error vector of tributary $B_n$ under one set of runoff data is expressed as $(Y^1, Y^2, \ldots, Y^J)$, and its error matrix can be expressed as $\left[ y^j_{n,i} \right]_{M \times J}$ where $n$, $n \in (1, 2, \cdots, N)$ is the tributary from which runoff is derived.

Figure 1 shows that there are many factors affecting the runoff supplied to downstream reservoirs, which could come from the main stream and tributaries. Thus, the inflow forecast errors are multivariate. Taking the second reservoir as an example, the inflow forecast error matrix of the main stream $A$ and tributary $B_1$ are $\left[ x_i^j \right]_{M \times J}$ and $\left[ y^j_{1,i} \right]_{M \times J}$, respectively. The inflow forecast errors of the second reservoir are then a combination of the error matrices of $A$ and $B_1$, which have $M \times M$ combinations in total. Correspondingly, the forecast errors of reservoir $k$ should be the combination of the error matrices of $A$, $B_1$, $B_2, \ldots, B_n$, with $M^{(n+1)}$ combinations in total.

*2.2. Stochastic Simulation of Multivariate Inflow Forecast Errors*

GMM is suitable for describing the inflow forecast errors with different distribution characteristics with its flexibility; thus it was used in this study. The distribution function of the inflow forecast error variable during one forecast period is one-dimensional. Assuming that the forecast error variable of inflow *A* is $X^j$, its probability density function can be obtained by using the GMM, such that:

$$f(x^j; \theta) = \sum_{k=1}^{K} \alpha_k N(x^j | u_k, \sigma_k^2) \tag{4}$$

where *K* is the number of mixed Gaussian models; $\theta$ is the parameter to be estimated in the model; $\alpha_k$, $u_k$ and $\sigma_k^2$ are respectively the weight, the mean value and the variance of the Gaussian distribution *k*, $\sum_{k=1}^{K} \alpha_k = 1$; $N(x^j | u_k, \sigma_k^2)$ is the expression of Gaussian distribution.

Similarly, the set of probability density functions of tributary $B_n$ is:

$$f(y_n) = \left\{ f\left(y_n^j\right) | j = 1, 2, \ldots, J \right\} \tag{5}$$

Kendall's rank correlation coefficient method is applicable to the calculation of the correlation coefficient between two or more rank variables. Therefore, the correlation coefficient $\tau$ between each forecast error variable can be calculated by the Kendall rank correlation coefficient method, as shown in Equations (6) and (7):

$$\tau = \frac{2}{n(n-1)} \sum_{1 \le i \le j \le n} sign[(x_i - x_j)(y_i - y_j)] \tag{6}$$

$$sign(x) = \begin{cases} 1 & x > 0 \\ 0 & x = 0 \\ -1 & x < 0 \end{cases} \tag{7}$$

A Copula function can describe the correlation between random variables [27,28]. The high-dimensional meta-student *t* Copula (*t*-Copula) and high-dimensional meta-Gaussian copula in the family of meta-elliptic Copula Functions are commonly used in hydrology. Compared with the high-dimensional Gaussian copula, high-dimensional *t*-Copula can describe tail correlation among variables. Because it is difficult to quantify the interference factors in the prediction process, the forecast error may deviate from the mean value at a certain time point. High-dimensional *t*-Copula can describe the characteristics that the forecast errors deviate greatly from the mean value at a certain time point. Therefore, based on the analysis of correlation coefficients among forecast error variables in the different forecast periods, with $f(x^j)$ and $f\left(y_n^j\right)$, a stochastic joint distribution model of *X* and *Y* can be established by using *t*-Copula. If the marginal distribution function of the forecast error variable of tributary *n* during the forecast period *t* is defined as $u_t^n$, then the joint distribution function of all runoff process forecast errors can be developed according to Equation (8):

$$C\left\{ \left(u_1^1, u_2^1, \ldots, u_J^1\right), \left(u_1^2, u_2^2, \ldots, u_J^2\right), \ldots \left(u_1^{(N+1)}, u_2^{(N+1)}, \ldots, u_J^{(N+1)}\right) \right\}$$
$$= t_{\Sigma, v}\left[ t_v^{-1}(u_1^1), \ldots, t_v^{-1}(u_J^{(N+1)}) \right] \tag{8}$$

where $C\{\cdot\}$ is copula function, $t_{\Sigma, v}(\cdot)$ is the *t*-distribution function with $v$ degrees of freedom, whose covariance matrix is $\sum$ and $t_v^{-1}(\cdot)$ is the inverse *t*-distribution function with $v$ degrees of freedom, and *N* is the total number of tributaries.

For the goodness of fit test, this study utilized the squared Euclidean distance ($D^2$). The $D^2$ value is calculated as follows:

$$D^2 = \sum_{i=1}^{n} \left| \widehat{C}_n(x_i, y_i) - \widehat{C}(x_i, y_i) \right|^2 \tag{9}$$

where $\widehat{C}_n(x_i, y_i)$ is the empirical copula function.

The steps involved in the simulation of the inflow series based on GMM-Copula are as follows:

Step 1: Establish the forecast error matrix using the measured and forecast runoff data.

Step 2: Use the appropriate probability density function to fit the forecast error variable. In this study, the GMM was adopted to fit the inflow forecast error function for each forecast period, the initial value of the GMM was determined by K-mean clustering analysis [29], and its applicability was tested with the K-S test.

Step 3: Analyze the correlation between the forecast error variables using the Kendall rank correlation coefficient method.

Step 4: Use a t-Copula function to fit the marginal distribution function of each forecast error variable and obtain the joint distribution function.

Step 5: Based on the developed joint distribution function, use the Monte Carlo method to simulate the forecast errors and obtain M sets of inflow forecast errors.

Step 6: Calculate the simulated inflow according to Equation (2).

## 3. Risk Analysis for Short-Term Operation of the Power Generation in Cascade Reservoirs

### 3.1. Short-Term Operation of the Power Generation Model of Cascade Reservoirs

3.1.1. Maximum Power Generation Capacity Model of Cascade Reservoirs

For the formulation of a power generation plan, the inflow is the forecast runoff process, and the maximum cascade power generation capacity is taken as the objective to establish the optimal scheduling model:

$$E = max \sum_{t=1}^{T} N_t \Delta t = max \sum_{t=1}^{T} \sum_{i=1}^{n} K Q_i^t H_i^t \Delta t \tag{10}$$

where $E$ is the total power generation capacity of the system; $n$ and $T$ are the total number of hydropower stations and total periods considered, respectively; $N_t$ is the total output of the system in the $T$ period; $\Delta t$ is the length of the period; $K$ is the output coefficient; $Q_i^t$ is the generation reference flow of $i$ power station in the $t$ period; and $H_i^t$ is the generating head of $i$ power station in the $t$ period.

3.1.2. Minimum Energy Consumption Model of Cascade Reservoirs

The principle of minimum energy consumption of the cascade hydropower stations is used for load distribution among hydropower stations to implement the power generation plan, and the objective function is as follows:

$$F = min \sum_{i=1}^{n} \sum_{t=1}^{T} H_i^t Q_i^t \tag{11}$$

where $F$ is the total energy consumption of the system; $n$ and $T$ are the total number of hydropower stations and total periods considered, respectively; $Q_i^t$ is the generation reference flow of $i$ power station in the $t$ period; and $H_i^t$ is the generating head of $i$ power station in the $t$ period.

### 3.1.3. Restrictions

The short-term optimal operation in cascade reservoirs is constrained by reservoir and hydropower station-related characteristics and comprehensive utilization rules related to water supply and power generation. These are outlined below.

Water volume balance constraint

$$V_i^{t+1} = V_i^t + (Q_{i-1}^{t-\tau_i} + A_{i-1}^{t-\tau_i} + R_i^t - Q_i^t - A_i^t)\Delta t \tag{12}$$

where $V_i^t$ and $V_i^{t+1}$ are the initial and final storage capacities of reservoir $i$ in period $t$, respectively; $\tau_i$ is the flow lag time of reservoir $i$; $R_i^t$ is the interval inflow of reservoir $i$ in period $t$; and $A_i^t$ is the abandoned water flow of reservoir $i$ in period $t$.

Reservoir capacity constraint

$$V_{i,t}^{min} \leq V_i^t \leq V_{i,t}^{max} \tag{13}$$

where $V_i^t$ is the water storage capacity of reservoir $i$ in period $t$, and $V_{i,t}^{min}$ and $V_{i,t}^{max}$ are the minimum and maximum water storage capacity of $i$ in period $t$, respectively. Generally, $V_{i,t}^{min}$ is the storage capacity corresponding to the dead water level, and $V_{i,t}^{max}$ is the storage capacity corresponding to the water level limit set for flood control during the flood season or the storage capacity corresponding to the normal pool level during the non-flood season.

Reservoir discharge constraint

$$q_i^{min} \leq q_i^t \leq q_i^{max} \tag{14}$$

where $q_i^t$ is the discharge flow of reservoir $i$ in period $t$, and $q_i^{min}$ and $q_i^{max}$ are the minimum and maximum allowable discharges from reservoir $i$, respectively.

Output constraint

$$N_{i,t}^{min} \leq N_i^t \leq N_{i,t}^{max} \tag{15}$$

where $N_{i,t}^{min}$ and $N_{i,t}^{max}$ are the minimum and maximum electrical output limits of the hydropower station $i$ in period $t$, respectively, and $N_i^t$ is the output of hydropower station $i$ in period $t$.

Power balance constraint

$$N_{i,t}^{plan} = N_{i,t}^{actual} \tag{16}$$

where $N_{i,t}^{plan}$ is the power generation of the hydropower station $i$ in period $t$ when the power generation plan is made, and $N_{i,t}^{actual}$ is the power generation of the hydropower station $i$ in period $t$ when the power generation plan is implemented.

Variables are not negatively constrained.

Due to the large number of reservoirs and the large scale of the calculation involved in the modeling of the short-term optimal operation of large-scale cascade reservoirs, a uniform self-organizing map genetic algorithm (UGA) [30] was used to run the power generation model.

### 3.2. Risk Analysis for Short-Term Operation of the Power Generation in the Cascade Reservoirs

The risk analysis for short-term operation of the power generation in the cascade reservoirs is mainly conducted to calculate the probability of breaking a constraint and to provide risk indicators for decision-makers. In the risk analysis of the operation of power generation, the usual risk events include the failure of hydropower generating units to deliver the scheduled output, the exceedance of maximum or minimum operating water level control ranges in the reservoir and the non-utilization of more water in the reservoir for power generation than was predetermined. The risk rate of power generation operation is the probability of occurrence of these risk events. In this paper, three indicators were used.

Insufficient output risk rate: the probability demonstrates that the hydroelectric generating set cannot produce the predetermined output. This is mathematically expressed as:

$$r_1 = P(N'_t < N^*_t) = \frac{m^1_p}{M} \times 100\% \tag{17}$$

where $N'_t$ is the actual output of the system in period $t$, $N^*_t$ is the predetermined output of the system in time period $T$, the predetermined output of each period is obtained by calculating the maximum power generation capacity model of cascade reservoirs, $M$ is the total simulated runoff, and $m^1_p$ is the number of damaged runoff processes. If the system output is blocked in one period of a runoff process, the runoff process is a damaged runoff process.

Beyond-or-below-limit water level risk rate: the probability of the water level exceeding the operating water level control range is mathematically expressed as:

$$r_2 = \frac{m^2_p}{M} \times 100\% \tag{18}$$

where $m^2_p$ is the number of damaged runoff processes. If the water level of any reservoir in any period of a runoff event exceeds the operating water level control range, the runoff is considered damaged.

Wasted water risk rate: the probability of wasted water at the cascade hydropower stations is mathematically expressed as follows:

$$r_3 = P\left(W_{t,a} > W^*_t\right) = \frac{m^3_p}{M} \times 100\% \tag{19}$$

where $W_{t,a}$ is the actual wasted water volume of the system in period $t$, $W^*_t$ is the planned wasted water quantity of the system in the period, and $m^3_p$ is the number of damaged runoff processes $t$. If there is a period of time when $W_{t,a}$ is more than $W^*_t$, then runoff is not fully utilized and is considered to be damaged runoff.

*3.3. Calculation Steps for Short-Term Operation of the Power Generation Risk Rate of Cascade Reservoirs*

The steps taken to calculate risks associated with short-term operation of the power generation of a cascade hydropower system include the following (Figure 2):

Step 1: Taking the forecast runoff process as the input, the maximum power generation model for cascade reservoirs is calculated. Then the power generation plan of the system is obtained.

Step 2: The runoff of the M field is simulated based on the stochastic simulation (Section 2.2), which includes multivariate inflow forecast errors.

Step 3: Using the runoff data and the power generation plan of the M field as input, the calculation for short-term operation of the power generation in the reservoir group is carried out to assess the minimum energy consumption of the cascade system.

Step 4: The risk rate of insufficient output, based on the above-or-below water level limit and amount of wasted water, is calculated.

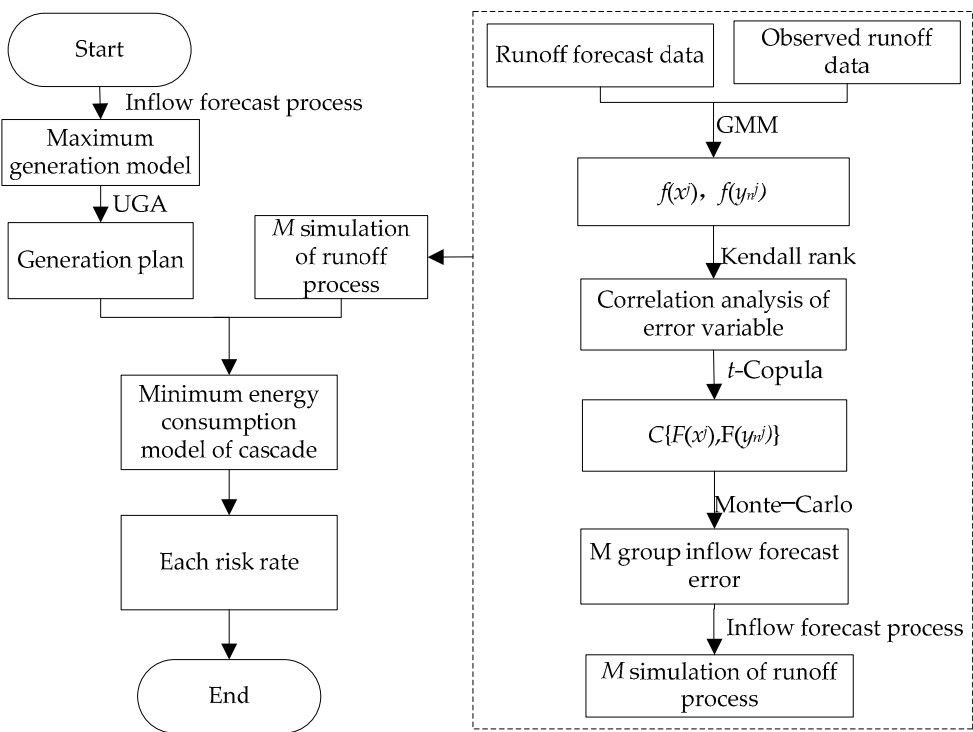

**Figure 2.** Steps involved in calculating the risk rate.

## 4. Case Study

To verify the methodology proposed in this paper, the Jinguan cascade, consisting of three hydropower stations (Jinxi, Jindong and Guandi) within the Yalong River basin, was used as a case study. Since the Jinguan hydropower stations were put into operation, the formulated power generation plans have been affected by inflow forecast errors, which has led to insufficient power generation or wasted water during some periods. The Jindong hydropower station is a diversion-type hydropower station. Inflow to the Guandi Reservoir includes the outflow from the Jindong Reservoir and the inflow from the Jiulonghe tributary, which enters the channel along the river segment between the Jindong Reservoir and Guandi Reservoir. During the flood season, flood control is a priority; thus, the runoff data during the dry season from 2013 to 2017 were selected to study the risk for short-term operation of the power generation. The spatial distribution of the hydropower stations is shown in Figure 3, and their general characteristics are listed in Table 1.

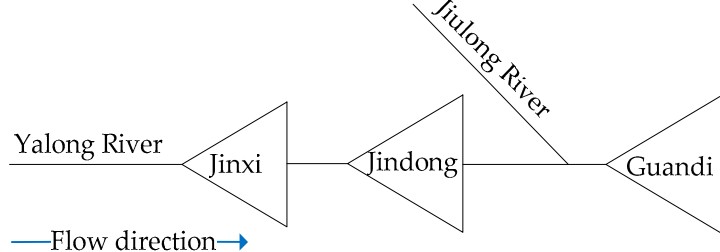

**Figure 3.** Schematic diagram of the Jinguan cascade hydropower system.

**Table 1.** Characteristics of the Jinguan cascade hydropower stations.

| Hydropower Station | Installed Capacity MW | Firm Power MW | Designed Annual Energy Output $\times 10^8$ kW·h |
|---|---|---|---|
| Jinxi | 3600 | 1086 | 166.20 |
| Jindong | 4800 | 1443 | 237.60 |
| Guandi | 2400 | 709.80 | 110.16 |

*4.1. Analysis of Forecast Error Characters in Different Forecast Periods*

Due to the limitations of forecast data, only 6, 12, 18 and 24 h inflow forecast data were available. Therefore, an interpolation method was used to obtain runoff data for the other forecast periods in a day. The length of a forecast period is one hour. Based on the analysis of the measured historical runoff data and the forecast data, the forecast error parameters of the Jinxi Reservoir inflow and Guandi Reservoir interval inflow (i.e., inflow to the reach between the reservoirs) in the different forecast periods were obtained by the GMM (Tables 2 and 3, respectively). The number of mixed Gaussian models and the initial parameter values of the GMM were determined according to Ji et al. [24]. According to reference [24], simplified treatment $K$ was taken as 2 for the Jinxi Reservoir inflow and Guandi Reservoir interval inflow.

**Table 2.** Fitted Gaussian mixture model (GMM) parameters of Jinxi Reservoir inflow in different forecast periods.

| Forecast Period | $K$ | Weight ($\alpha$) | Mean Value ($u$) | Variance ($\sigma^2$) |
|---|---|---|---|---|
| 6 h | 2 | $\alpha_1 = 0.9577, \alpha_2 = 0.0423$ | $u_1 = -1.0001, u_2 = 18.8304$ | $\sigma_1^2 = 32.5671, \sigma_2^2 = 28.6849$ |
| 12 h | 2 | $\alpha_1 = 0.4928, \alpha_2 = 0.5072$ | $u_1 = -3.0380, u_2 = -0.8373$ | $\sigma_1^2 = 112.2997, \sigma_2^2 = 29.8075$ |
| 18 h | 2 | $\alpha_1 = 0.6912, \alpha_2 = 0.3088$ | $u_1 = 1.6888, u_2 = -2.5080$ | $\sigma_1^2 = 50.7381, \sigma_2^2 = 36.5538$ |
| 24 h | 2 | $\alpha_1 = 0.6404, \alpha_2 = 0.3596$ | $u_1 = -2.8084, u_2 = 3.6344$ | $\sigma_1^2 = 47.6983, \sigma_2^2 = 78.0387$ |

**Table 3.** Fitted GMM parameters of Guandi Reservoir interval inflow in different forecast periods.

| Forecast Period | $K$ | Weight ($\alpha$) | Mean Value ($u$) | Variance ($\sigma^2$) |
|---|---|---|---|---|
| 6 h | 2 | $\alpha_1 = 0.8264, \alpha_2 = 0.1736$ | $u_1 = -0.2519, u_2 = 0.5758$ | $\sigma_1^2 = 10.8288, \sigma_2^2 = 0.3925$ |
| 12 h | 2 | $\alpha_1 = 0.2239, \alpha_2 = 0.7761$ | $u_1 = -0.2722, u_2 = -0.5653$ | $\sigma_1^2 = 50.2290, \sigma_2^2 = 7.8424$ |
| 18 h | 2 | $\alpha_1 = 0.6840, \alpha_2 = 0.3160$ | $u_1 = -0.4602, u_2 = 1.7341$ | $\sigma_1^2 = 15.5119, \sigma_2^2 = 50.1798$ |
| 24 h | 2 | $\alpha_1 = 0.4690, \alpha_2 = 0.5310$ | $u_1 = -0.5955, u_2 = -0.1940$ | $\sigma_1^2 = 69.3037, \sigma_2^2 = 12.8721$ |

From Tables 2 and 3 we can see that there were no trends or similarities of each parameter with respect to the increase of periods due to random generation of errors.

Figure 4 shows the performance of the single Gaussian distribution model (GM) and the GMM in fitting the inflow forecast errors of the Jinxi Reservoir inflow and Guandi Reservoir interval inflow for the different forecast periods. From the perspective of describing the overall characteristics of the error data, the GMM fit the data better than that of the single Gaussian distribution. The fitting curve of the GMM was closer to the empirical data, which means that the GMM is better than the GM in describing the probability density estimation. Thus, the GMM was employed to create the inflow forecast error probability density function in each forecast period.

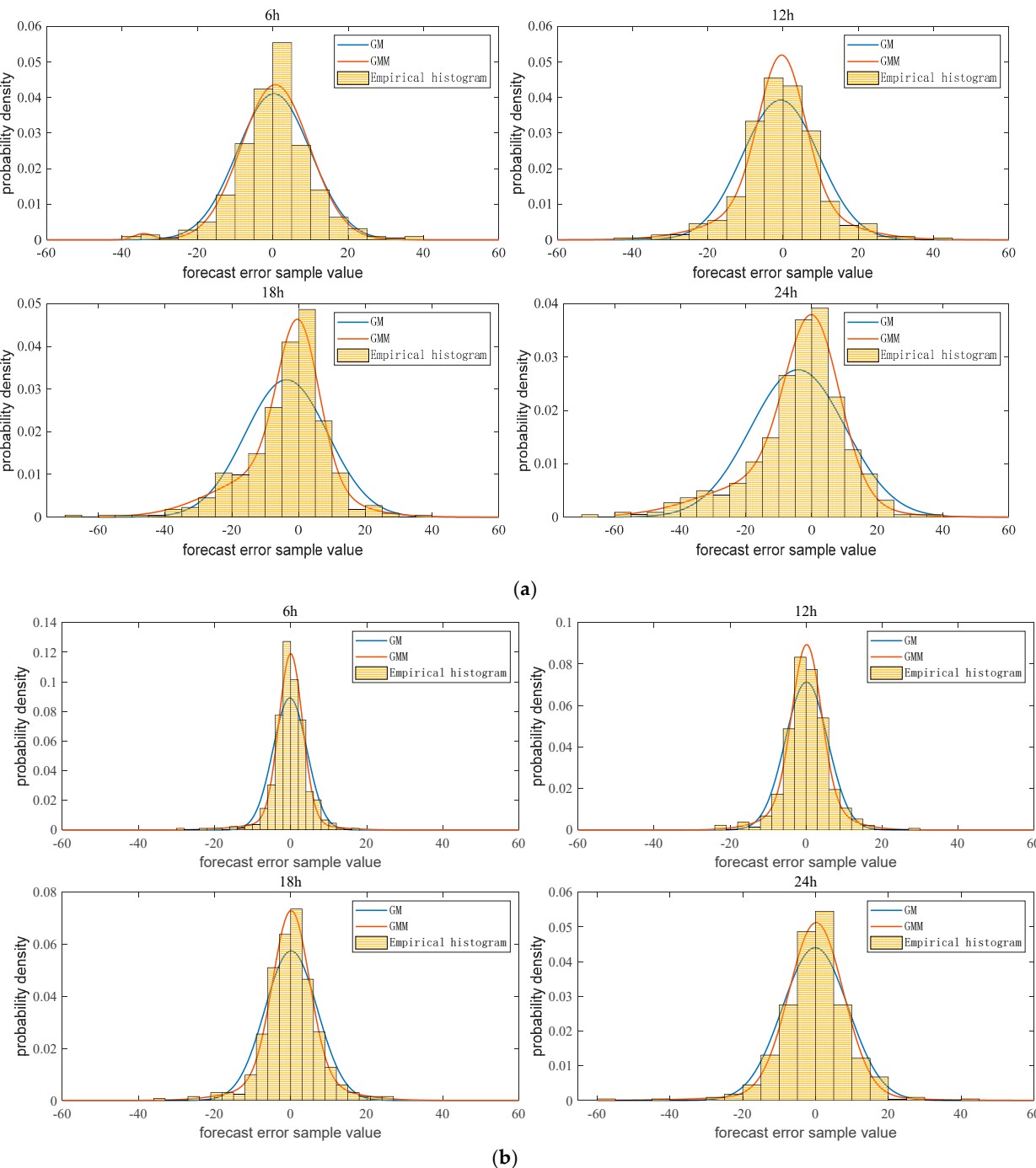

**Figure 4.** Probability density function curves of inflow forecast errors; (**a**) Jinxi Reservoir inflow; (**b**) Guandi Reservoir interval inflow.

Figure 5 illustrates the probability density function curves of inflow forecast errors of the Jinxi Reservoir inflow and the Guandi Reservoir interval inflow in different forecast periods as determined by the GMM. Figure 5 shows that with an increase in the forecast period, the shape of the error distribution gradually changes from sharp and narrow to short and wide, indicating that forecast uncertainty rises with an increase in time. This pattern conforms to the general expectations of deterministic forecast models; for inflow along the Guandi interval, the inflow forecast error probability density function curves exhibit a larger difference among each forecast period. This is also expected because there are many uncertainty factors affecting the inflow to the reach between the reservoirs,

making it difficult to determine the inflow from the upper reaches of the Jiulong River tributary. Consequently, the longer the forecast period, the greater the forecast errors.

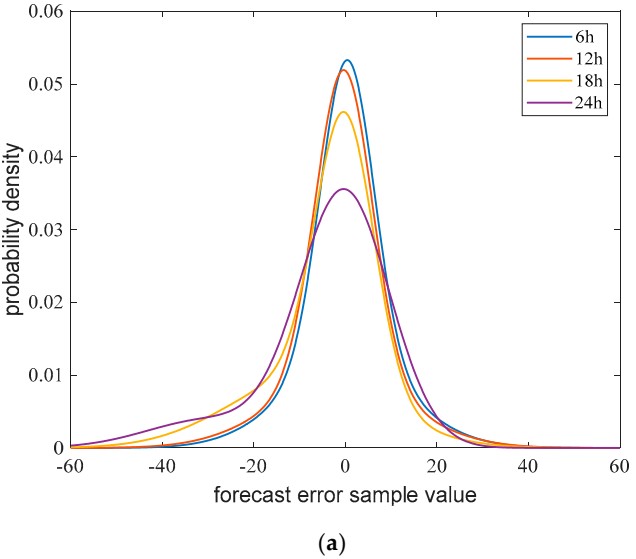

(**a**)

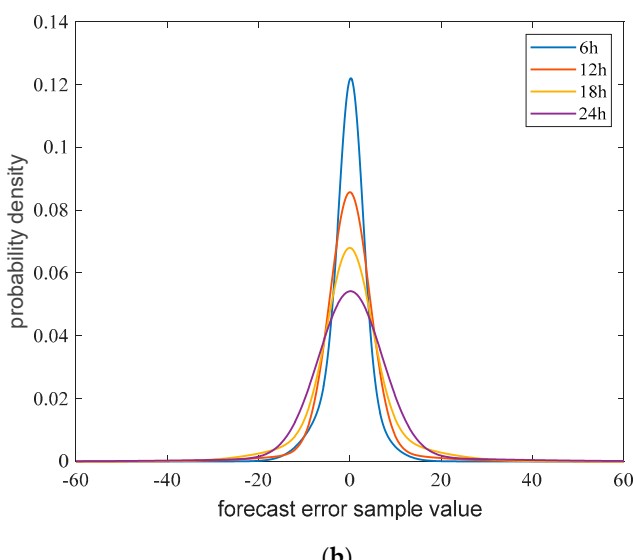

(**b**)

**Figure 5.** Probability density function curves of inflow forecast errors in different forecast periods; (**a**) Jinxi Reservoir inflow; (**b**) Guandi Reservoir interval inflow.

### 4.2. Analysis of the Joint Distribution Function

The correlation between each forecast period's inflow errors was initially analyzed, where $x(6)$, $x(12)$, $x(18)$ and $x(24)$ are the respective inflow forecast errors for the Jinxi Reservoir inflow for the 6 h, 12 h, 18 h and 24 h forecast periods. The parameters $y(6)$, $y(12)$, $y(18)$ and $y(24)$ are the respective inflow forecast errors of the Guandi Reservoir interval inflow for the 6 h, 12 h, 18 h and 24 h forecast periods. The Kendall correlation coefficient was used to determine the relationships between the parameters of the differing forecast periods, and a bilateral test was carried out. The bilateral significant level was 0.05. Table 4 shows the calculation and test results.

**Table 4.** Correlation coefficients of inflow forecast errors between each forecast period.

| | | $x_{(6)}$ | $x_{(12)}$ | $x_{(18)}$ | $x_{(24)}$ | $y_{(6)}$ | $y_{(12)}$ | $y_{(18)}$ | $y_{(24)}$ |
|---|---|---|---|---|---|---|---|---|---|
| $x_{(6)}$ | Correlation coefficient | 1.000 | 0.312 | 0.123 | 0.230 | −0.022 | 0.032 | −0.030 | 0.020 |
| | *p* value of bilateral significance test | - | 0 | 0 | 0 | 0 | 0 | 0 | 0 |
| $x_{(12)}$ | Correlation coefficient | 0.312 | 1.000 | 0.160 | 0.226 | 0.001 | 0.001 | 0.003 | −0.016 |
| | *p* value of bilateral significance test | 0 | - | 0 | 0 | 0 | 0 | 0 | 0 |
| $x_{(18)}$ | Correlation coefficient | 0.123 | 0.160 | 1.000 | 0.282 | 0.021 | 0.003 | −0.040 | 0.001 |
| | *p* value of bilateral significance test | 0 | 0 | - | 0 | 0 | 0 | 0 | 0 |
| $x_{(24)}$ | Correlation coefficient | 0.230 | 0.226 | 0.282 | 1.000 | −0.004 | −0.005 | −0.011 | 0.008 |
| | *p* value of bilateral significance test | 0 | 0 | 0 | - | 0 | 0 | 0 | 0 |
| $y_{(6)}$ | Correlation coefficient | −0.022 | 0.001 | 0.021 | −0.004 | 1.000 | −0.025 | 0.004 | 0.034 |
| | *p* value of bilateral significance test | 0 | 0 | 0 | 0 | - | 0 | 0 | 0 |
| $y_{12}$ | Correlation coefficient | 0.032 | 0.001 | 0.003 | −0.005 | −0.025 | 1.000 | 0.006 | 0.023 |
| | *p* value of bilateral significance test | 0 | 0 | 0 | 0 | 0 | - | 0 | 0 |
| $y_{(18)}$ | Correlation coefficient | −0.030 | 0.003 | −0.040 | −0.011 | 0.004 | 0.006 | 1.000 | 0.001 |
| | *p* value of bilateral significance test | 0 | 0 | 0 | 0 | 0 | 0 | - | 0 |
| $y_{(24)}$ | Correlation coefficient | 0.020 | −0.016 | 0.001 | 0.008 | 0.034 | 0.023 | 0.001 | 1.000 |
| | *p* value of bilateral significance test | 0 | 0 | 0 | 0 | 0 | 0 | 0 | - |

Table 4 demonstrates that the *p* values of the two-sided hypothesis test are less than the significance level. Therefore, the zero hypothesis is rejected. There is a correlation between the forecast errors of each prediction time, and the forecast errors of each forecast time are correlated.

The *t*-Copula functions were used to fit the forecast errors joint distribution function of the Jinxi Reservoir inflow and Guandi Reservoir interval inflow in each forecast period. The $D^2$ value of the *t*-Copula was 0.1658. The fitting result is good. Hence, the *t*-Copula function was selected to fit the forecast errors of the Jinxi Reservoir inflow and the Guandi Reservoir interval inflow in each forecast period.

Based on the stochastic simulation model of inflow forecast errors using GMM-Copula (GMM and *t*-Copula), 10,000 sets of inflow forecast errors of the Jinxi Reservoir inflow and the Guandi Reservoir interval inflow were simulated (Table 5).

**Table 5.** Parameter values of the simulated and measured errors.

| Forecast Period | Inflow | Mean Value | | Variation Coefficient | | Variance | |
|---|---|---|---|---|---|---|---|
| | | Simulated | Measured | Simulated | Measured | Simulated | Measured |
| 6 h | Jinxi Reservoir inflow | 0.315 | 0.309 | 30.573 | 31.456 | 96.504 | 94.453 |
| | Guandi Reservoir interval inflow | −0.034 | −0.033 | −114.425 | −116.529 | 21.141 | 20.659 |
| 12 h | Jinxi Reservoir inflow | −0.646 | −0.669 | −14.778 | −15.179 | 104.810 | 103.239 |
| | Guandi Reservoir interval inflow | −0.031 | −0.030 | −195.442 | −196.682 | 33.812 | 34.822 |
| 18 h | Jinxi Reservoir inflow | −3.733 | −3.626 | −3.291 | −3.420 | 149.759 | 153.795 |
| | Guandi Reservoir interval inflow | 0.493 | 0.508 | 15.009 | 14.582 | 55.801 | 54.902 |
| 24 h | Jinxi Reservoir inflow | −4.229 | −4.133 | −3.602 | −3.495 | 210.848 | 208.739 |
| | Guandi Reservoir interval inflow | 0.642 | 0.661 | 14.050 | 13.544 | 79.093 | 80.054 |

From Table 5 we can see that, when using the GMM-Copula, the calculation results were similar to the measured values. The results show that the random simulated sequence has nice simulation effect, since it keeps the correlation, overall distribution and error-specific distribution of the original sequence. The results show that the random simulated sequence can keep the correlation, overall distribution and error-specific distribution of the original sequence; the simulation effect is good. Therefore, the random simulated sequence can be used for risk analysis.

### 4.3. Risk Analysis for Short-Term Operation of the Power Generation

Based on the simulated forecast errors and the actual Jinxi Reservoir inflow and Guandi Reservoir interval inflow data, 10,000 sets of simulated reservoir inflow data were obtained for short-term operation of the power generation. A wet day, normal day and dry day during the non-flood season from 2013 to 2017 were selected to calculate the risk indexes and analyze their patterns in variation. The resulting risk index values of the cascade reservoirs' short-term operation of the power generation are provided in Table 6. According to the power generation plan, if the water level falls below the lower limit, the power output will be undermined; when the water level exceeds the upper limit, there will be wasted water. The risk rate represents the sum of the risk rate of the beyond-upper-limit water level and the risk rate of the below-lower-limit water level.

**Table 6.** Risk index value.

| Typical Day | Risk Rate of Insufficient Output/% | Risk Rate of Wasted Water/% | Risk Rate of Beyond- or-Below-Limit Water Level/% |
|---|---|---|---|
| Wet day | 1.48 | 2.02 | 3.50 |
| Normal day | 1.56 | 0.63 | 2.19 |
| Dry day | 1.80 | 0.52 | 2.32 |
| Mean value | 1.61 | 1.06 | 2.67 |

The following information can be extracted from Table 6. The risk rate of insufficient output differs between typical days. For the wet day, the risk rate of insufficient output was the lowest because runoff was abundant, whereas for the dry day, due to less runoff, the inflow was fully utilized in the formulated power generation plan. The forecast accuracy had a considerable influence on the actual output, resulting in the highest risk rate of insufficient output; the risk rate of wasted water also differed between typical days. It was the highest during the wet day and the lowest for the dry day. For the wet day when runoff was abundant, the power generation plan was formulated with the forecast inflow, and wasted water was reduced as much as possible to maximize power generation. However, during the actual operation (which used the actual inflow), due to less inflow forecast error, the risk rate of the beyond-upper-limit water level was relatively high. The risk rate of the beyond-or-below-limit water level was the highest on the wet day and the lowest on the normal day. For the normal day, the risk rates of both insufficient output and wasted water were comparatively small, thereby producing the lowest risk rate for the beyond-or-below-limit water level.

### 5. Discussion

1. From the analysis of this paper, it can be found that the simulated result is close to the measured data, and the average value of simulated accuracy is 97.52%. The average value of simulated accuracy for reference [25] is 97.87%. Compared with reference [25], the simulated accuracy is similar. The GMM-Copula in this study also exhibited a satisfactory performance that was consistent with those reported by Ji et al. [24]. These indicate that the methodology proposed in this study can effectively describe the statistical characteristics of the inflow forecast error series and

provides a reference value for short-term operation of the power generation in large cascade reservoirs.

2.  From the aspect of risk rate, the hydropower generation plan usually takes the forecast runoff as the input data directly, without considering the inflow forecast errors, which leads to the risk and failure of the generation plan. We also can see there is little difference in the risk rate value of insufficient output among the three representative days. The predetermined output we select is the target output obtained with the forecast runoff process as the input, while the calculated output is obtained with the simulated runoff process based on the forecast error as the input. Because the forecast error distribution function we consider is the same, the risk rate value is similar. The risk rate value obtained in this study can be a useful reference for the decision-making. Therefore, this paper not only considers the risk of hydropower generation caused by inflow forecast errors but also analyzes and discusses the seasonal change of risk of hydropower generation in different periods including the dry period and wet period in detail. Compared with reference [26] which also analyzes the risk of short-term operation of the power generation of reservoirs, the simulated forecast errors considering the correlation between each forecast period are closer to the actual process, and the risk rate value in our study can be more reasonable. When planning hydropower generation, it is helpful to reduce the risk rate of power generation by adding the prediction value and the simulation prediction errors.

3.  This approach provides some guidance for hydropower station operations. Since the characteristics of the inflow forecast error vary seasonally, the risk rate of hydropower generation operation also varies. Therefore, examination of the risk for hydropower generation operation of cascade reservoirs under inflow uncertainty for different runoff periods for a comprehensive analysis will be the focus in the next study.

## 6. Conclusions

The forecast error variables of reservoir inflow and interval inflow were jointly fit using a GMM-Copula, which considered the correlation between inflow sources. The stochastic model of inflow forecast errors in multiple forecast periods was then established. The simulated runoff was used to analyze the risk of short-term operation of the power generation. The analysis found that:

1.  GMM-Copula model was more suitable to simulate the inflow errors in different forecast periods. By comparing the mean values, variance and variation coefficients of the simulated and the actual inflow forecast errors, the accuracy of the joint simulation was greater. Thus, the proposed approach provides a novel means of simulating inflow forecast errors with multivariate combinations.

2.  Through the analysis of power generation risk during the non-flood season, it was determined that the risk rates of wasted water (3.50%) and beyond-or-below-limit water levels (2.02%) were the highest on wet days. The risk rate of insufficient output was the highest on dry days, which offers new insights into the short-term operation of the power generation of the Jinguan hydropower stations.

**Author Contributions:** Conceptualization, Y.W. (Yueqiu Wu) and Y.W. (Yi Wang); methodology, L.W.; validation, Y.Z., J.W. and Q.M.; formal analysis, Y.W. (Yueqiu Wu) and X.L.; investigation, Y.W. (Yi Wang); resources, Y.W. (Yueqiu Wu) and L.W.; writing—original draft preparation, Y.W. (Yueqiu Wu) and Y.W. (Yi Wang); writing—review and editing, Y.W. (Yi Wang); visualization, Y.Z. and B.H.; supervision, J.W.; project administration, Y.W. (Yueqiu Wu). All authors have read and agreed to the published version of the manuscript.

**Funding:** This study was funded by the National Natural Science Foundation of China (Grant number: 51709105), the Thousand Talent Program for Young Outstanding Scientists of China (Grant number: Y771071001), Fundamental Research Funds for the Central Universities (Grant number: 2020MS026), China Postdoctoral Science Foundation (Grant number: 2020M680487) and the Guangdong Foundation for Program of Science and Technology Research (Grant number: 2020B1111530001).

**Institutional Review Board Statement:** Not applicable.

**Informed Consent Statement:** Not applicable.

**Data Availability Statement:** Not applicate.

**Conflicts of Interest:** The authors declare no conflict of interest.

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
