# Peer review of "Risk Analysis for Short-Term Operation of the Power Generation in Cascade Reservoirs Considering Multivariate Reservoir Inflow Forecast Errors"

_sustainability, doi:10.3390/su13073689_

Round 1

Reviewer 1 Report

  1. Overview and general recommendation:

The authors of the article have undertaken an interesting topic.

In order to solve the problem, the inflow runoff forecast error variables were jointly fitted using a t Copula function based on the analysis of the error distribution characteristics in different forecast  periods.

The proposed solution was validated   using  the  Jinguan  cascade hydropower system within the Yalong River basin as a case study.

Unquestionably, the topic of involving the use of t Copula function to the analysis of the error distribution characteristics in different forecast  periods has been known for many years and has been used with great success on an industrial scale. The results of such works have already been published in many international journals.

Nevertheless, the authors of the article have proposes a novel methodology  uses the t Copula function based model to the analysis of the error distribution characteristics in different forecast  periods to  analyze  the  risk  of  insufficient  power  generation including in particular in the short-term power generation  operation of  cascade reservoirs.

The presented material corresponds to the profile of the Journal " Sustainability ". The scientific value of the submitted material qualifies the article for publication in this Journal. The article may be published after completing and correcting all issues. I recommend that a minor revision is necessary. I made the detailed comments in point 2.1. I ask that the authors specifically address each of my comments in their response.

2.1. Major comments:

  1. Section 5: Discussion

Please clarify and expand on this point. The description is too general and does not refer directly to the analyzes performed, including the presented "Case Study".

  1. Section 7: The conclusions formulated correspond with the presented research results, but they should be redefined and clarified, as they should display elements of novelty in the context of using the results of research and analysis. In this situation, the article presents an interesting research methodology, which, however, does not bring revolutionary elements in the context of innovative methods and technologies in scientific research in the subject area of the undertaken research problem. Therefore, it is recommended to clearly indicate the novelty of the proposed solution.

To sum up, the "Discussion" and "Conclusions" should be formulated in such a way as to present the key results obtained in effect of the completed research using proprietary methods.

2.2. Minor comments:

In the context of the above-mentioned comments, the current point 5 ("Discussion) should be changed to" Results and discussion ".

Author Response

Point 1. Section 5: Discussion

Please clarify and expand on this point. The description is too general and does not refer directly to the analyzes performed, including the presented "Case Study".

Response: Thanks for your advice. The discussion part has carried on the detailed analysis combined with the "Case Study". At present, the discussion is carried out from three aspects, the first one is the comparative analysis of simulation error and actual error, the second one is from the aspect of risk rate, and the last one is the next step should be done under the current research problems. The following is a revised discussion:

(1)From the analysis of this paper it can be found that the simulated result is close to the actual data, . the maximum and the minimum accuracy of the simulation is 99.37% and 96.26%, respectively, which indicating that the methodology proposed in this study can effectively describe the statistical characteristics of the runoff forecast error series and   provides an effective basis for short-term power generation operation of large cascade reservoirs.

(2)From the aspect of risk ratio, the hydropower generation plan usually takes the forecast runoff as the input data directly, without considering the runoff forecast error, which leads to the risk and failure of the generation plan. So this paper not only considers the risk of hydropower generation caused by runoff forecast error, but also analyzes and discusses the change of risk of hydropower generation in different periods including dry period and wet period in details. Specifically, for the wet day, the risk ratio of insufficient output was the lowest because runoff was abundant, whereas for the dry day, the inflow was fully utilized in the formulated power generation plan. The forecast accuracy had a considerable influence on the actual output, resulting in the highest risk ratio of insufficient output. The risk ratio of wasted water also differed between typical days. It was the highest during the wet day and the lowest for the dry day. For the wet day when runoff was abundant, the power generation plan was formulated with the forecast inflow and wasted water was reduced as much as possible to maximize power generation. However, during the actual operation (which used the actual inflow), due to the runoff forecast error, the risk ratio of the beyond-upper-limit water level was relatively high. The risk ratio of the beyond-or-below-limit water level was the highest in the wet day and the lowest in the normal day. For the normal day, the risk ratio of both insufficient output and wasted water were comparatively small, thereby producing the lowest risk ratio for the beyond-or-below-limit water level.So when plan the hydropower generation, it is helpful to reduce the risk rate of power generation by adding the prediction value and the simulation prediction error.

(3) This approach provides some guidance for hydropower station operations. Since the characteristics of the runoff forecast error varies seasonally, so the risk ratio of hydropower generation operation also varies. Therefore, examine the risk of hydropower generation operation of cascade reservoirs under runoff uncertainty for different runoff periods for a comprehensive analysis will be the focus in the next study.

Point 2. Section 7: The conclusions formulated correspond with the presented research results, but they should be redefined and clarified, as they should display elements of novelty in the context of using the results of research and analysis. In this situation, the article presents an interesting research methodology, which, however, does not bring revolutionary elements in the context of innovative methods and technologies in scientific research in the subject area of the undertaken research problem. Therefore, it is recommended to clearly indicate the novelty of the proposed solution.

To sum up, the "Discussion" and "Conclusions" should be formulated in such a way as to present the key results obtained in effect of the completed research using proprietary methods.

Response: Thanks for your advice. The conclusion has been revised to highlight the innovation work. Previous studies mostly consider the runoff forecast error of a single reservoir and rarely consider the correlation of forecast error between different sources of runoff processes. Few studies have examined the risk of power generation operation of cascade reservoirs caused by forecast error of different runoff processes. This study proposes a novel approach to do so.

Point 3.  2.2. Minor comments:

In the context of the above-mentioned comments, the current point 5 ("Discussion) should be changed to" Results and discussion ".

Response: Thanks for your advice. The current point 5 ("Discussion) has been changed to" Results and discussion ". And the corresponding modification has been carried out.

Reviewer 2 Report

Dear the authors, I am pleased to review your manuscript, entitled “Risk analysis of short-term power generation operation of cascade reservoirs considering multivariate reservoir inflow runoff forecast error,” and possibly to help improve the manuscript based on my review comments. The detailed comments to the authors are attached as a PDF. Thank you,

Author Response

Overall comments

The authors seemed to talk about the risk analysis that comes from the impacts of the inflow-forecast errors on the power generation operations within short periods (up to a day), based on my understanding of this manuscript (MS). An uncertainty-based model was implemented to a relatively, complicated river system that consists of two reservoirs and one tributary as a case study. However, there are insufficient explanations and skipped definitions of words (maybe, technical terms) in many places. Therefore, I was unable to understand exact methodology. Because the current form of this MS is still inappropriate for the publication, I highly recommend that the authors should improve the MS.

Response: Thanks for your valuable advice, those constructive comments and suggestions will be very helpful to improve the quality of the manuscript. We have modified the content accordingly which are highlighted in red color in the revised manuscript, and the detailed responses for each comment is shown in below for your kind review.

Major comments

Point 1. I agree with the authors’ idea that considering uncertainties in runoff simulations and multiple periods is important to evaluate accurate risks related to power generation operations. However, a part of methods in this study is greatly similar to that of the existed study by Ma et al. (2021) that I found recently. Especially, the authors mentioned that “multivariate runoff forecast error” is a key in this study. However, there is no citation of Ma’s study in the MS. Please make sure what is the difference between your study and Ma’s study. I guess that introducing t-Copula and tributaries with Gaussian mixture model (GMM) might be different from Ma’s study. The Introduction section does not talk about the differences from past studies except for multiple reservoirs’ implementation. Please make the differences clearer and enhance the uniqueness of this study in the Introduction section.

Response: Thanks for your kind advice, and I very impressive to your erudition and conscientious. The article you have referred was published in January 2021. Our article was submitted on January 15, 2021. So at that time, there was no way to refer Ma's paper. Now, according to your suggestion, we have added the reference of Ma's article in the introduction of our manuscript. Both our and Ma's article have put the consideration on runoff forecast error, while the difference is that Ma's paper focuses on the improvement of GMM method and establishes IGMD model to simulate the inflow error of a reservoir. But this paper focuses on the analysis of the relationship between the runoff forecast errors of the main stream and tributaries of the basin. The GMM and t-copula models are used to simulate runoff forecast errors, and the risk ratio of short-term hydropower generation operation of cascade reservoirs caused by the forecast errors is calculated. Therefore, the research objectives and focus of these two articles are different.

The differences from previous studies have been listed in the introduction l59-l63, as follows.

“Previous studies mostly consider the runoff forecast error of a single reservoir and rarely consider the correlation of forecast error between different sources of runoff processes. Few studies have examined the risk of power generation operation of cascade reservoirs caused by forecast error of different runoff processes. This paper proposes a novel approach to do so.”

Point 2. The discussion section in the MS is very poor. Please extend the section with deep insight discussions/suggestions and comparable studies. Plus, I think that authors’ implications for the values in some tables e.g., Table 4) are not scientific but are likely to be their beliefs. Please check the implications of results.

Response: Thanks for your kind advice. According to your suggestion, we have analyzed and discussed the research method and the results in detail in the discussion part. Specifically, there are three points. (1) Through the comparative analysis of the simulated inflow forecast error and the actual inflow forecast error, it shows that the proposed model and method can effectively describe the statistical characteristics of the runoff forecast error series, and provide an useful reference for the short-term power generation operation planning of large cascade reservoirs. (2) From the aspect of risk ratio, the hydropower generation plan usually takes the forecast runoff as the input data directly, without considering the runoff forecast error, which leads to the risk and failure of the generation plan. So this paper not only considers the risk of hydropower generation caused by runoff forecast error, but also analyzes and discusses the change of risk of hydropower generation in different periods including dry period and wet period in details. (3) On the basis of the current research works, the future research plan is also put forward. Moreover, can the correlation of inflow forecast error between different forecast periods can be determined by calculating the correlation coefficient of them, which can provide the basis for fitting the marginal distribution function with t-copula function. Therefore, the correlation coefficients of inflow forecast errors between different forecast periods shown in Table 4 are the basis of the mathematical calculation of this study.

Point3: GMM was applied to this study. However, Ma et al. (2021) suggest that the GMM has at least two problems: the difficulty of determining the mixture number of Gaussian distributions and the effect of the initialization for GMM parameters on iteration results, based on past studies (e.g., Ji et al.). Why did the authors dare to employ the GMM?

Response: Thanks for your kind advice. Each method has its advantages and disadvantages. Although GMM has few disadvantages in some certain cases, but the structure of GMM is clear and  flexible, so it has good applicability to describe the inflow runoff forecast error with different distribution characteristics. In this paper, to study the forecast error of different inflow forecast periods, GMM was more suitable. Through the comparative analysis of the simulation error and the actual error in the example analysis, it can be seen that the simulation accuracy of GMM and t-copula is higher. For the mixture number k of Gaussian distribution, this paper referred paper as: [Ji, C.; Liang, X.; Zhang, Y.; Liu, Y. Stochastic model of reservoir runoff forecast errors and its application. J. Hydroelectr. Eng. 2019, 38, 75–85.].

The parameter “k” was simplified and value 2 was selected accordinly. For parameter initialization, this paper used K-means,. the references for this method had been cited in l123-124. Due to the limited time, IGMD and t-copula are not combined to fit the distribution function of inflow forecast error of cascade reservoirs. This method will be further studied in the future.

Point4:  Is it new to introduce a ‘t-Copula’ function that can combine reservoir-related multi-variables, including multi-forecast periods, and tributary-related variables with GMM? However, the t-Copula is one of the conventional methods. Please clearly explain what a notable method or a new idea is in this study through the whole sentences.

Response: Thanks for your kind advice. Although t-copula method is commonly used in hydrological analysis, it is rare to combine GMM with t-copula to analyze the relationship between reservoir related variables (including multiple prediction periods) and tributary related variables, which is an innovation of this paper.

Point 5: Many unclear, insufficient explanations, some undefined symbols/words, several mis-choice words, grammar errors, and some Chinese characters appear in this MS. Please check all the sentences and words, including figures and tables from the basic level.

References

Ma et al. (2021) Quantifying risk propagated from inflow forecast 2 uncertainty to reservoir operation coupling with flood 3 and electricity curtailment risk. Water, 13(2), 173.

Ji, C.; Liang, X.; Zhang, Y.; Liu, Y. Stochastic model of reservoir runoff forecast errors and its application. J. Hydroelectr. Eng. 2019, 38, 75–85.

Response: Thanks for your advice. I have checked all the sentences and words, including figures and tables from the basic level.

Minor comments

Abstract:

Point 6: L11-L12 “The factors causes …” → I think it is a grammar error. Please check it.

L11 What is an exact meaning of “inflow runoff”? I usually see runoff, inflow, and outflow in hydrological processes. Please make sure if the phrase is general.

Response: Thanks for your advice.  L11-L12 “The factors causes …”has been changed into “The factors cause …”.L11 “inflow runoff” should be “inflow”, and I have corrected it.

Point 7: L14-L15 “The inflow runoff forecast error of different forecast periods may also be correlated” →What is correlated to? Is it the effect on inflow?

Response: Thanks for your advice. This correlation means that there is a certain relationship between the inflow forecast errors in different forecast periods. And it is the effect on inflow. We simulate the inflow forecast error through the inflow prediction error function to guide the operation. Without considering the correlation between them, the simulated t-copula function is not too accurate.

Point 8: In L11~L15 The authors mentioned that the inflow is affected by two factors (the error from the upstream reservoir and the error caused by tributaries). Please make sure what the authors wanted to mention from the sentences in L11 - L15.

Response: Thanks for your advice. When considering the runoff forecast error, the inflows of cascade reservoirs are usually calculated separately, and then superimposed. The correlation between inflow forecast error of upstream reservoir and interval inflow forecast error is not considered. In this study, t-copula was used to establish the joint distribution function of different inflow and different forecast time.

Point 9:  L16 “in this paper” → “in this study”

Response: Thanks for your advice. Yes, it should be “in this study”。And I have corrected it.

Point 10: “A stochastic model of inflow runoff forecast error in multiple forecast periods” → What does this mean? Does it mean “a stochastic model that may produce inflow runoff forecast errors in multiple forecast periods?”

Response: Thanks for your advice. The stochastic model was based on the joint distribution function of inflow forecast error obtained by t-copula function, and used Monte Carlo stochastic simulation method to simulate the inflow forecast error of each reservoir in each forecast period.

Point 11: L18-L21 Please separate one sentence into two sentences because the sentence is long and unclear.

Response: Thanks for your advice. L18-L21 I have separated one sentence into two sentences.

Point 12: L21 “the approach” → What kind of approach did the authors intend? The authors mentioned only the description of a stochastic model, forecast error, t-Copula etc. as method tools. Please clarify the approach as a method process or action.

Response: Thanks for your advice. “the approach”is “To address this multivariate problem, the inflow forecast error variables were jointly fitted in this study using GMM and a t-Copula function based on the analysis of the error distribution characteristics in different forecast periods. A stochastic model of inflow runoff forecast error in multiple forecast periods was established. This model can be used to simulate inflow runoff, and analyze the risk of insufficient power generation, wasted water and above-or-below-reservoir limit water levels in the short-term power generation operation of cascade reservoirs.”

Point 13: L22 “The risk of its …” and “simulation of its inflow” → What do both “its” mean? Are they “the Yalong River basin”? If so, here is one example of changing the sentence. For this case study, the risk analysis for short-term operation of the power generation was performed …

Response: Thanks for your advice. They are “the Yalong River basin”. And the sentence has been changed as “For this case study, the risk analysis for short-term operation of the power generation was performed based on stochastic simulation of the inflow forecast error.”

Point 14: L24 “the results verified” → “the results showed” is better. Note that “verify the results” is grammatically correct.

Response: Thanks for your advice. L24 “the results verified”has been changed into “the results showed”.

Point 15: L24 “the proposed methods” → The authors did not mention what methods they employed. The authors talked only about the stochastic model, forecast error, t-Copula etc. I do not know how the authors used the model with other elements. This unclear explanation is the same as “approach”. Please clarify this.

Response: Thanks for your advice. The method proposed in this study was to deal with the simulation prediction error. Firstly, GMM method was used to fit the probability density function of each forecast period, then t-copula was used to fit the probability density function of each forecast period of the main stream and the interval inflow, and finally Monte Carlo method was used to simulate the prediction error. The details can be seen in abstract L19-L21.

1.Introduction

Point 16: L29 “the operation of short-term power generation” → I feel confused. Which one is correct, “short-term operation of the power generation” or “operation of the short-term power generation”? If the latter is correct, please clearly define “short-term power generation” because this phrase seems to be uncommon.

Response: Thanks for your advice. “the operation of short-term power generation” has been changed into “short-term operation of the power generation”.

Point 17: L 35 “because of deviations between the actual and predicted inflow” → This phrase can be placed in the end of the sentence because your sentence is hard to read.

Response: Thanks for your advice. This phrase has been placed in the end of the sentence.

Point 18: L37 “currently be directly applied to” → It is strange. It could be changed into ‘be currently applied directly to’

Response: Thanks for your advice. “currently be directly applied to” has been changed into “be currently applied directly to”.

Point 19: L38 “power production” → For an appropriate expression, is it ‘power generation processes’ because the authors mentioned that “the schemes” cannot be applied to?

Response: Thanks for your advice. The schemes can be applied to power generation processes. However, due to the existence of prediction error, the risk rate of insufficient power generation will be caused. This study was to analyze these risk rates.

Point 21: L38 “the study of the risk of short-term” → It is strange. Is it ‘risk analyses for short-term …’ as the authors mentioned in abstract?

Response: Thanks for your advice. L38 “the study of the risk of short-term” should be  “the risk analysis of short-term”. And I have checked it in the study.

Point 22: L47 “reduces the simulation accuracy of runoff forecast error.” → It is unclear. Is it ‘reduces the accuracy by means of the multiple errors (from different periods)’? Please clarify what the authors wanted to mention.

Response: Thanks for your advice. It is “reduces the accuracy by means of the multiple errors (from different periods)”. And I have corrected it in the study.

Point 23: L48 “To analyze the influence of forecast errors for different forecast periods on forecast error” →This phrase is the same as ‘the influence of errors on error.’ What is a difference between the first error and the second error?

Response: Thanks for your advice. The first error is “forecast errors for different forecast periods”. The second error is “runoff process forecast errors”. I have checked it in the study.

Point 24: L51 “which are often in basins” → ‘which are usually placed in basins’ is better.

Response: Thanks for your advice. I have changed “which are often in basins” into “which are usually placed in basins”.

Point 25: L57 to 60 “Previous studies mostly … rarely …”, “Few studies …”→ It is necessary to add representative papers as a citation.

Response: Thanks for your advice. I have added representative papers.

Point 26: L61 “to do so” → Please write it exactly as an appropriate academic writing.

Response: Thanks for your advice. I have changed int into “to consider the correlation between all forecast errors in the forecast period”.

Point 27: L61 “the forecast error variable function” → What does this mean? Please explain it shortly if it is a new idea. Or please add the citation for the function if it is not new.

Response: Thanks for your advice. It is not new, and I have added the citation.

Point 28: L63 “the forecast error variables” → What kind of variables did the authors use although “variables” often appear in the section of Introduction?

Response: Thanks for your advice. The error of a forecast period is a variable. The error function of different forecast periods is different.

Point 29: L65 “an array of forecasted runoff” → What does it mean?

Response: Thanks for your advice. “an array of forecasted runoff”should be“an array of simulated inflow”. “simulated inflow” is the simulation of actual inflow. We can get the simulated inflow according to the Equation (8) in the paper. 

  1. Materials and Methods

Point 30: L69 “each” → Is it ‘each tributary’? The sentence in L68-69 is unclear.

Response: Thanks for your advice. Yes, it is “each tributary”. I have checked it in L68-69.

Point 31: L69 “Some of these forecast errors may be related while others may not.” → This sentence is very rough. It is hard to understand what the authors mentioned. For example, what are some errors related to?

Response: Thanks for your advice. This refers to the correlation between these prediction errors. “Some of these forecast errors may be related while others may not.”has been changed into “There may be correlation between some of these prediction errors while others may not.”

2.1. Multivariate runoff forecast error

Point 32: L76 “historical forecast runoff data” → What does this mean? Are the data the reproducibly simulated data by a runoff model based on historical events? Do the authors want to mention the relative runoff forecast error, defined with forecast data and actual (observed) data, like Equation (1)? Pleas clarify it.

Response: Thanks for your advice. “historical forecast runoff data” is “forecast inflow”. I have checked it in the study.

Point 33: Figure 1 → Which direction does the river water flow, right to left or left to right? Please add an arrow on the figure. Plus, why did the authors assume that only one tributary exists in each path between reservoirs? Please explain this assumption.

Response: Thanks for your advice. The direction of the river water flow is from left to right. And I have added an arrow on the figure. In order to simplify the calculation, we assume that only one tributary exists in each path between reservoirs.

Point 34: L89 “Thus, the runoff forecast error is multivariate.”→ The authors assumed that errors come from multivariate factors. What kind of factors did the authors pick up?

Response: Thanks for your advice. Multivariate factors are prediction errors of main stream and tributaries.

Point 35: L93 “Correspondingly, the forecast error of reservoir k …” → Why is “n” assigned like B n and M (n +1) .  Equation (3) → What is the symbol, N(. |., .)? Please define it clearly.

Response: Thanks for your advice. The runoff forecast error of the second reservoir is then a combination of the error matrices of A and B1, which has M×M combinations in total. Correspondingly, the forecast error of reservoir k should be the combination of the error matrices of A, B1, B2Bn, with  combinations in total. N(. |., .)is the expression of Gaussian distribution and has been indicated in the study.

2.2. Stochastic simulation of multivariate runoff forecast error

Point 36: L98 “using the Gaussian Mixture Model (GMM)” → Why did the authors use this method? Ma et al. (2021) suggested that an advanced model of GMM (exactly speaking, Gaussian Mixture Distribution, GMD), named IGMD, is better than one of the conventional models, GMD. Please explain the advantage of this method (GMD) clearly if compared with IGMD.

Response: Thanks for your advice. Each method has its own advantages and disadvantages. GMM is simple in structure and flexible in shape, which is suitable for describing the inflow forecast errors with different distribution characteristics. In this study, the forecast error of different inflow forecast periods is studied, and GMM is more suitable. Through the comparative analysis of the simulation error and the actual error in the example analysis, it could be seen that the simulation accuracy of GMM and t-copula was higher. Due to the time limit, there was no detailed study on IGMD, which will be carried out in the future.

Point 37: L104 “by the Kendall rank correlation coefficient method” → Why was ‘Kendall ... method’ used in this study? Please explain the reason shortly here.

Response: Thanks for your advice. There are many methods to calculate the correlation coefficient, but this study only used Kendall rank correlation coefficient method.

Point 38: Equation (7) → What are “C{}” and “L”? What is “N”? This symbol causes the confusion with N(,) in Equation (3).

Response: Thanks for your advice. “C{}”is copula function. “L” should be “…”, and I have corrected it. “N”is Gauss function.

Point 39: L116-L128 → Bullet-use explanation is good for readers, but the numbering (1) to (6) should be changed into difference symbols because the same symbols are used in the numbering of equations. Please change them.

Response: Thanks for your advice. I have changed them.

Point 40: L120 “its applicability was tested” → What was the GMM applied to in some tests?

Response: Thanks for your advice. K-S test is used in this study. K-S test has been added in this paper.

Point 41: L123 “t-Copula” → why did the authors employ the t-Copula with the assumption of Student's t-distribution, instead of ‘Gaussian Copula.’ This question is the same as the introduction of “t-distribution function” in Equation (7).

Response: Thanks for your advice. Because it is difficult to quantify the interference factors in the forecast model, the forecast error may deviate from the mean value at a certain time point, and the high-dimensional t-copula does not have tail correlation. Therefore, on the premise of deriving the error distribution of each forecast time, this study selects t-copula to establish the joint distribution of inflow forecast errors of multiple forecast times.

Point 42: Equation (8) → What is the difference between “forecast” and “simulated”? Does Q sumilated mean the inflow (discharge), involving the error between forecast inflow and observed inflow? If so, please add more information for Q sumilated . Note that for Ma et al (2021)’s paper, they defined it as a corrected forecast error.

Response: Thanks for your advice. Q sumilated means the inflow (discharge), involving the error between forecast inflow and observed inflow. where  is the corrected forecast inflow;  is the forecast inflow; Xj is the simulated forecast error. And I have defined in the study.

  1. Risk analysis of short-term power generation operation of cascade reservoirs

3.1. Short-term power generation operation model of cascade reservoirs

3.1.1. Maximum power generation model of cascade reservoirs

Point 43: L133-L134 “the … power generation is taken as the objective.” → This sentence does not make sense. That is, the power generation is not equal to an objective. Please make sure if your grammars are correct for all sentences. Is “power generation” electric-generating capacity? This is because power generation is a process but is not a quantity.

Response: Thanks for your advice. “the … power generation is taken as the objective.” Should be “the … power generation capacity is taken as the objective.” And I have corrected it in the study.

Point 44: Equation 9 → Please explain what some symbols (e.g., E, K, H t i ) are just after the equation. I know that these symbols are defined in the next sub-sub section. However, this manner is strange.

Response: Thanks for your advice. I have added the explanation of the symbols after the Equation (9).

3.1.2. Minimum energy consumption model of cascade reservoirs

3.1.3. Restrictions

Point 45: L146 “comprehensive utilization constraints” → What does this mean?

Response: Thanks for your advice. The restriction of comprehensive utilization includes water supply and power generation,and so on.

Point46: L148 “(1) Water balance constraint” → Is it ‘Water volume balance …’?

Response: Thanks for your advice. “Water balance constraint” is “Water volume balance …”. And I have changed into “Water volume balance …”.

Point 47: L177 “the problem” → What problem did the authors want to mention? What did “the large number of … and the large scale of …” cause?

Response: Thanks for your advice. The problem is the optimal operation model of cascade reservoirs. “the large number of … and the large scale of …” cause dimension disaster. It takes a long time to solve. Sometimes it is even difficult to find the optimal solution.

Point 48: Equations (14) and (15) → What is “N”? This symbol is messed up with the same symbol in Equations (3) and (7).

Response: Thanks for your advice. “N”is  the system output in Equations (14), (15), (3) ,(7). To prevent confusion, we have changed “N”in Equations (14) as .

3.2. Short-term power generation operation risk analysis of cascade reservoirs

Point 49: L179 “The short-term power generation operation risk analysis” → This is a long phrase. Please make it clearly with a preposition.

Response: Thanks for your advice. “The short-term power generation operation risk analysis”has been changed into “The short-term operation of power generation risk analysis”.

Point 50: L182 “the usual risk events are the failure of” → This sentence is the same as “the events are the failure.” What does this mean? Event and failure are not consistent.

Response: Thanks for your advice. L182 “the usual risk events are the failure of”should be “the usual risk events includes the failure of”.

Point 51: L188-L200 The symbols ((1)~(3)) in bullets can be changed because the same symbols (numbering) are used in equations.

Response: Thanks for your advice. These symbols have been changed.

Point 52: L188-L189 “The probability that the hydroelectric generating set cannot produce the predetermined output” → Is this sentence correct? I guess this is a grammar error.

Response: Thanks for your advice. This sentence is correct.

Point 53: Equation (16) → I am not clear about this equation. This is because the authors mentioned that the set cannot produce the predetermined output (N * t ). How was the predetermined output determined?

Response: Thanks for your advice. The predetermined output is based on the generation plan. When the hydropower station is in actual operation, the generation plan should be specified first.

Point 54:Equation (17) → Is “M” the same as “M” in Equation (18)?

Response: Thanks for your advice. “M”in Equation (17) is the same in Equation (18).

3.3. Calculation steps of short-term power generation operation risk rate of cascade reservoirs

Point 55: L210 “after which the power generation plan of the system is obtained;” → It is unclear.

Response: Thanks for your advice. This place should be “Taking the forecast runoff process as the input, the maximum power generation model for cascade reservoirs is calculated. Then the power generation plan of the system is obtained.”

  1. Case Study

Point 56: L228 “the formulated power generation plans have been skewed by runoff forecast error,” → What does this mean? Does this mean that the forecast error affected the discharge (or water level) that can be used in the plans? Please clarify this sentence.

Response: Thanks for your advice. This means that the forecast error affected the discharge (or water level) that can be used in the plans.

Point 57: Table 1 → What does “Regulation performance” mean? Why does a big gap between Jinxi and the others exist in time scale?

Response: Thanks for your advice. The regulation performance of reservoir is the capacity of regulating runoff and the degree of meeting the load demand of power grid. The regulation performance of the reservoir has been determined during the construction of the reservoir.

Point 58: Figure 3 → Please add the arrow as a flow direction.

Response: Thanks for your advice. I have added the arrow as a flow direction.

4.1. Analysis of error characters in different forecast periods

Point 59: L248-L250 “The enumeration method showed that the most suitable value of the Gaussian mixture parameter K was 2 for the Jinxi Reservoir inflow and Guandi Reservoir interval inflow.” It is necessary add the reference that shows K = 2 as the best choice.

Response: Thanks for your advice. I have added the reference [24] that shows K = 2 as the best choice.

Point 60: Tables 2 and 3 → Where is “μ” defined in the MS? The mean value in GMM was defined as “u” in Equation 3. Please make the symbol consistent. Plus, the parameters such as weighted and mean value in the errors are greatly variated among forecast periods. Does this mean that there might be no trends or similarities of each parameter with respect to the increase of periods due to random generation of errors?

Response: Thanks for your advice. The sign of the mean value has been unified as u in the MS.

Point 61: L259 “the GMM is better than the GM in describing the probability density estimation” → Is it correct that the GM has a single Gaussian distribution? Please clarify it.

Response: Thanks for your advice. GM is a single Gaussian distribution model. And it was added in the MS.

Point 62: Figure 5a → The distributions are asymmetry. Especially, probability densities in the negative x-axis range are slightly higher than those in the positive range. Why did the distributions appear?

Response: Thanks for your advice. GM is asymmetric, GMM is symmetric. This shows that GMM has a good fitting effect.

4.2. Analysis of the joint distribution function

Point 63: L286 “The correlation between …” → Were Equations (5) and (6) used to compute the correlation?

Response: Thanks for your advice. Yes, Equations (5) and (6) were used to compute the correlation.

Point 64: Table 5 → How did the authors define “actual error”?

Response: Thanks for your advice. The actual prediction error is calculated by Equation (1).

Point 65: L297-L302 “Table 4 shows …” → “(1) a relatively strong correlation” Does it mean 0.315 between x(6) and x(12), 0.282 between x(18) and x(24), etc.? Do these values indicate a string correlation?

Response: Thanks for your advice. The correlation value is the correlation between the forecast errors of the two forecast periods.

Point 66: “(2) some correlation between inflow …” I cannot see the correlation between x(.) and y(.). These cross-correlation values are very low.

Response: Thanks for your advice. These cross-correlation values are very low. Not equal to 0 means there is a certain correlation.

Point 67: “(3) it is feasible to use …” I cannot understand this conclusion at all. Why did the authors think that?

Response: Thanks for your advice. If there is correlation between them, t-copula can be used to find their joint distribution function. So this sentence is right.

Point 68:L306 “L-distance method” → What is this method? There is no explanation of it.

Response: Thanks for your advice. The L-distance method is the Euclidean distance between the selected copula function and the empirical copula function.

Point 69:L307 “The D value” → What is it?

Response: Thanks for your advice. “The D value” is the size of L-distance.

4.3. Risk analysis of short-term power generation operation

Point 70: Table 6 and related sentences (L334-L347) → The authors mentioned that lowest or maximum values in each risk rate. However, most of these values indicate similarly low percentage. Are there any significant findings among these small values when you comparing among three days (wet, normal, and dry)? If the small differences are important, please clearly explain how important the effects of the small differences on power generation operation are.

Response: Thanks for your advice. When making power generation plan, it is helpful to reduce the risk rate of power generation by adding the prediction value and the simulation prediction error.

5 Discussion and 6. Conclusions

Point 71: I am sorry that I will not look further at sentences one by one in both sections due to my time limitation. Plus, I think that these sections will be changed dramatically in a future revised MS because the current form of MS seems to be immature for an academic journal publication. Again, I highly recommend that the authors should improve the MS carefully with sufficient, appropriate explanations and accurate, scientific writings.

Response: Thanks for your advice. The full text has been revised according to the comments.

References

Point 72: L396 “1. Zhang, J.Y. “ → Is this paper written in Chinese? Please make the citation correctly and exactly.

Response: Thanks for your advice. The references have been revised.

Reviewer 3 Report

Line 38 formatting mistake [5]

Figure 2 formatting mistake in algorithm between GMM - Kendal

Conclusions will be more scientific if the risk values from table No.6 will be added into conclusions in lines 375-379. Example "it was determined that the risk rates of wasted water (2.02%) and beyond-or-below-limit (3.05%) water levels were the highest on wet days".

A uniform self-organizing map genetic algorithm (UGA) was used to solve the problem in this paper, I would recommend for authors to continue research in future and to compare results obtained in this research with application of using another algorithms: simulated annealing algorithm (SA) or ant colony algorithm (ACO). Some scholars have done further research on these intelligent algorithms (Cui DW and Jin B 2016. Application of dynamic adaptive particle swarm optimization algorithm and least square support machine in annual runoff prediction. People's Pearl River   37(10) pp 27-33), and improved the operation mechanism of them, which provides effective methods and experience or the study of optimal scheduling of hydropower stations (Min Yi, Li Mo, and Qin Shen "Study on Generation Scheduling of Cascade Hydropower Stations Based on SAPSO," Journal of Coastal Research 104(sp1), 371-378, (5 October 2020). https://doi.org/10.2112/JCR-SI104-066.1)

Author Response

Point 1. Line 38 formatting mistake [5].

Response: Thanks for your advice. Line 38 formatting mistake has been corrected.

Point 2. Figure 2 formatting mistake in algorithm between GMM – Kendal.

Response: Thanks for your advice. Figure 2 formatting mistake in algorithm between GMM – Kendal has been corrected.

Point 3. Conclusions will be more scientific if the risk values from table No.6 will be added into conclusions in lines 375-379. Example "it was determined that the risk rates of wasted water (2.02%) and beyond-or-below-limit (3.05%) water levels were the highest on wet days".

Response: Thanks for your advice. The risk values from table No.6 have been added into conclusions in lines 375-379.

Point 4. A uniform self-organizing map genetic algorithm (UGA) was used to solve the problem in this paper, I would recommend for authors to continue research in future and to compare results obtained in this research with application of using another algorithms: simulated annealing algorithm (SA) or ant colony algorithm (ACO). Some scholars have done further research on these intelligent algorithms (Cui DW and Jin B 2016. Application of dynamic adaptive particle swarm optimization algorithm and least square support machine in annual runoff prediction. People's Pearl River 37(10) pp 27-33), and improved the operation mechanism of them, which provides effective methods and experience or the study of optimal scheduling of hydropower stations (Min Yi, Li Mo, and Qin Shen "Study on Generation Scheduling of Cascade Hydropower Stations Based on SAPSO," Journal of Coastal Research 104(sp1), 371-378, (5 October 2020). https://doi.org/10.2112/JCR-SI104-066.1)

Response: Thanks for your advice. Due to the limited time, this study only uses uniform self-organizing map genetic algorithm (UGA) to solve the problem, and not compared with other optimization algorithms. In the future work, other optimization algorithms will be studied and compared with each other.

Round 2

Reviewer 2 Report

Dear Authors, I made comment lists with PDF. Please watch my PDF as the attachment.

Author Response

Dear reviewer,

Thank you very much for reviewing our manuscript carefully and we are very much grateful for your efforts and time to provide us valuable comments and suggestions to improve the quality of our manuscript.

Please kindly see the attachment.

Round 3

Reviewer 2 Report

Dear the authors, I have reviewed your manuscript carefully and seriously. Please find my detailed comments in the attachment. Thank you,

Author Response

Dear reviewer,

Thank you very much for reviewing our manuscript again. We have carefully considered those comments and reply them one by one (please see “The third response” in below), and modified the manuscript accordingly. 
